# Heart Failure Impacts Endothelial Cell Responses to Cardiac Surgery on Cardiopulmonary Bypass

**DOI:** 10.3390/cells14171357

**Published:** 2025-08-31

**Authors:** Olga Papazisi, Rudmer J. Postma, Richard J. Dirven, Saskia L. M. A. Beeres, Remco R. Berendsen, Sesmu M. Arbous, Robert J. M. Klautz, Marieke E. van Vessem, Roel Bijkerk, Jan H. N. Lindeman, Meindert Palmen, Anton Jan van Zonneveld

**Affiliations:** 1Department of Cardiothoracic Surgery, Leiden University Medical Center, 2300 RC Leiden, The Netherlands; o.papazisi@lumc.nl (O.P.); r.j.m.klautz@lumc.nl (R.J.M.K.); m.palmen@lumc.nl (M.P.); 2Department of Internal Medicine, Division of Nephrology, Leiden University Medical Center, 2300 RC Leiden, The Netherlands; r.bijkerk@lumc.nl (R.B.); a.j.van_zonneveld@lumc.nl (A.J.v.Z.); 3Department of Internal Medicine, Division of Thrombosis and Hemostasis, Leiden University Medical Center, 2300 RC Leiden, The Netherlands; r.j.dirven@lumc.nl; 4Department of Cardiology, Leiden University Medical Center, 2300 RC Leiden, The Netherlands; s.l.m.a.beeres@lumc.nl; 5Department of Anesthesiology, Leiden University Medical Center, 2300 RC Leiden, The Netherlands; r.r.berendsen@lumc.nl; 6Department of Intensive Care, Leiden University Medical Center, 2300 RC Leiden, The Netherlands; m.s.arbous@lumc.nl; 7Department of Cardiothoracic Surgery, Amsterdam University Medical Center, 1105 AZ Amsterdam, The Netherlands; 8Department of Sports Medicine, Maxima Medical Center, 5504 DB Veldhoven, The Netherlands; 9Department of Vascular Surgery, Leiden University Medical Center, 2300 RC Leiden, The Netherlands; j.h.n.lindeman@lumc.nl

**Keywords:** heart failure, endothelial cells, postoperative hemodynamic instability, vasoplegia, cardiac surgery, cardiopulmonary bypass, mitochondria

## Abstract

Patients with heart failure with a reduced ejection fraction (HFrEF) are at an increased risk of developing postoperative hemodynamic instability and vasoplegia after surgery on cardiopulmonary bypass (CPB). Potentially pre-existing endothelial cell (EC) alterations due to chronic HF influence EC responses to cardiac surgery and might be responsible for the altered vascular responsiveness observed postoperatively. In this study, well-described EC activation markers were measured in blood samples collected pre- and perioperatively at four time points from HFrEF and control patients undergoing cardiac surgery on cardiopulmonary bypass (CPB). Circulating levels of Angiopoietin 2 (ANG2), von Willebrand Factor (vWF), and soluble P-selectin were measured using ELISA. Additionally, we investigated the responses of the cultured EC to patient-derived plasma through morphological profiling and mitochondrial functional assays. In total, 36 patients were included (67 (61–71) years, 78% male). HFrEF patients had higher baseline ANG2 and vWF levels when compared to controls. Both markers peaked during the first postoperative day. A pronounced increase in vWF was seen in controls after CPB. Ex vivo EC responses to patient-derived plasma showed distinct morphological differences between the two groups at baseline. A mitochondrial analysis indicated alterations in function and morphology for both groups after CPB. In conclusion, HFrEF patients exhibit a dampened EC response to cardiac surgery on CPB. Stable circulating factors in HFrEF plasma are responsible for inducing EC stress. Moreover, the mitochondrial function is highly affected postoperatively. This pre-existing mitochondrial and EC dysfunction predispose HFrEF patients to postoperative hemodynamic instability.

## 1. Introduction

Heart failure with reduced ejection fraction (HFrEF) is a clinical syndrome in which the heart fails to pump blood commensurate to the oxygen requirements of the body [1]. HFrEF is associated with high morbidity and mortality, poor quality of life, and high healthcare costs [2]. The global disease burden of HF in general is high, with a prevalence of 1–3% in developed countries, and it is expected to rise due to the aging of the population and better treatment [2,3]. Surgical treatments, such as restrictive mitral valve annuloplasty, surgical ventricular restoration, left ventricular assist device implantation, and, ultimately, cardiac transplantation, remain an important treatment option for end-stage chronic HFrEF patients not responding to optimal medical therapy or device therapy [4,5,6]. However, HF surgery on cardiopulmonary bypass (CPB) is associated with substantial perioperative morbidity, and HFrEF patients are at an elevated risk for developing postoperative vasoplegia [7,8,9,10]. Postoperative vasoplegia is characterized by low systemic vascular resistance due to the loss of the vascular tone and the blunted response to exogenous vasoconstrictors resulting in hypotension and end-organ malperfusion despite a normal or increased cardiac output, and is associated with poor clinical outcomes and prolonged hospitalizations [8,11,12,13].

HFrEF is associated with systemic endothelial cell (EC) dysfunction [14] resulting from increases in reactive oxygen species and the release of damage-associated molecular patterns (DAMPs) and matrix metalloproteinases (MMPs) from damaged cardiomyocytes or from post-ischemic damage and necrosis [15,16]. This pre-existing EC dysfunction might make ECs from HFrEF patients hypo- or hyperreactive to the major insults caused by surgery on CPB, ultimately resulting in vasoplegia. Various triggers have been suggested for this postoperative loss of vascular tone. Exposure of the blood to the foreign surfaces of CPB might trigger complement activation and systemic inflammatory responses, resulting in elevated levels of various EC agonists including thrombin, C5a, and the cytokines interleukin (IL)-1β, and tumor necrosis factor alpha (TNF-α) [17]. This systemic inflammatory response combined with the shedding of the EC glycocalyx caused by surgical trauma and reperfusion injury likely results in an even further activation of the endothelium [18,19,20].

Here, we aimed to study the EC responses of HFrEF patients and non-HF patients to surgery on CPB. We hypothesize that the ECs of HFrEF patients show different responses to surgery on CPB compared to non-HF patients undergoing similar surgeries, and that circulating plasma factors are partly responsible for this difference.

## 2. Material and Methods

### 2.1. Study Design

This study is an analysis of K2EDTA blood samples collected during the “Vasoresponsiveness in patients with heart failure” study; VASOR. VASOR is a prospective, observational study, conducted at Leiden University Medical Center. Patients with and without HFrEF undergoing cardiac surgery on CPB were compared. The study is registered at The Netherlands Trial Register (NTR5647) and was performed in line with the principles of the Declaration of Helsinki. All patients gave informed written consent prior to the inclusion in the study. The institutional medical ethical committee approved the protocol (22 May 2015, P14.298). The protocol was published online on November 2019 in the Journal of Cardiothoracic Surgery [21]. The study was reported according to the Strengthening the Reporting of Observational Studies in Epidemiology (STROBE) statement [22].

### 2.2. Blood Sample Collection

EDTA plasma samples were obtained at 4 different time points: T1, before induction of anesthesia; T2, after induction of anesthesia; T3, after CPB cessation; and T4, on day 1 postoperative. Whole-blood K2EDTA samples were centrifuged at 1550× *g* for 10 min at 4 °C to separate the blood plasma. Subsequently, plasma was transferred, aliquoted, and stored at −80 °C until further analysis.

### 2.3. Circulating Plasma Marker Levels

ANG2 levels were measured using a commercially available Solid Phase Sandwich ELISA kit (R&D Systems, Minneapolis, MN, USA; SANG-20 + WA126) according to the manufacture’s instruction. Plasma vWF levels were determined using an inhouse-developed Solid Phase Sandwich ELISA, according to de Jong et al. [23]. EDTA plasma samples were diluted 1:800, 1:1600, and 1:3200 in PBS + 0.1% Tween-20 before measuring by ELISA. CD138 levels were determined using a commercially available Solid-Phase Sandwich ELISA set (DIACLONE, Besancon, France; 851.620.005) according to the manufacturer’s instructions. Soluble P-Selectin (sP-selectin) levels were measured using a commercially available Solid-Phase Sandwich ELISA set (R&D Systems, Minneapolis, MN, USA; DY137). EDTA plasma samples were first diluted 1:50, 1:100, and 1:200 in PBS (Gibco, Waltham, MA, USA; 10010023) + 1% BSA (Sigma, St. Louis, MO, USA; A7030) before measuring by ELISA.

### 2.4. Cell Culture

Human Umbilical Vein ECs (HUVECs) were isolated non-invasively from umbilical cords discarded after birth at the Leiden University Medical Center, anonymized and under full consent of the parents, as previously described [24]. HUVECs were cultured in Endothelial Cell Growth Medium 2 (EGM-2) (PromoCell C-22111, Heidelberg, Germany) supplemented with 1% Penicillin/Streptomycin (Gibco 15070-063) for one passage and cryogenically frozen. HUVECs were thawed and cultured in a 1%-gelatin (Merck, Rahway, NJ, USA; 104078)-coated T75 flask for two days before transferring to a 1%-gelatin-coated, thin-bottom, 96-well flask (PerkinElmer Phenoplate, 6055302, Waltham, MA, USA). Cells were grown for 4 days prior to morphological profiling to form a confluent monolayer.

### 2.5. Ex Vivo EC Morphological Profiling

ECs were exposed to 25% plasma from HF or control patients at time point 1 (baseline) and 3 (directly after cessation of CPB) for morphological profiling, modified from Postma et al. [25]. Time point 1 and 3 were chosen, as these time points best represented baseline and the circulating factors released during surgery. Then, 0.5 µM recombinant Hirudin (ABCAM, Cambridge, UK; ab201396) and 25 µg/mL Corn Trypsin Inhibitor (Haematologic Technologies, Essex Junction, VT, USA; CTI-01) were added to K_2_EDTA plasma equaling 25% of the total volume. Basal Medium (EBM) (PromoCell C-22211) containing 1:100 insulin transferrin selenium supplement (ITS) (Gibco 41400) and 0.85 mM calcium-dichloride (CaCl_2_) (Merck 1.02381) was added to obtain the final volume [26].

Cells were incubated with MitoTracker Deep Red FM (Invitrogen M22426, Waltham, MA, USA) 1:2000 for 30 min, fixed by incubation with 4% PFA (Alfa Aesar J61899, Haverhill, MA, USA) for 10 min, and blocked for unspecific binding by blocking buffer (2% (*w*/*v*) BSA + 0.5% Glycine (Merck 1.04201) + 1% (*v*/*v*) Triton X-100 in PBS) for 30 min. Fixed cells were incubated for two hours with primary antibodies against vWF (, 1:1000, Aligent, DAKO a0082, Santa Clara, CA, USA) and VE-Cadherin (BD 555661, 2 µg/mL, BD Biosciences, Franklin Lakes, NJ, USA) in PBS with 2% BSA. Wells were washed with PBS + 2% BSA, followed by washing with 50 mM Borate Buffer, pH = 8.3. Subsequently, wells were incubated with 488 Alexa-fluorophore-labeled Donkey anti-mouse antibody (Invitrogen A-11001, 2 µg/mL), Qdot-625-labeled Goat anti-rabbit antibody (Invitrogen A-10194, 1:400), 1:1000 rhodamine phalloidin (Invitrogen R415), and 1:1000 HOECHST 33258 (Molecular Probes, Thermo Fisher Scientific, Waltham, MA, USA) in 50 mM Borate Buffer, at pH = 8.3 with 2% BSA + 0.05% glycine for one hour. Cells were washed three times with Borate Buffer + 2% BSA + 0.05% glycine, and stored under Borate Buffer.

Max-projections of 7 z-steps with 0.7 µm step size were acquired using a high content imager (Molecular Devices, ImageXpress™ Micro Confocal, San Jose, CA, USA) at 20× magnification with 60 µm pinhole, imaging six sites.

Morphological profiles were computed from the multiplexes using R (version 4.1.1) [27], using the R-package EBImage (version 4.29.2) [28] and inhouse scripts [29]. Cells were segmented by Voronoi-tessellation using nuclei as seeds and the VE-Cadherin staining to trace the borders.

### 2.6. Mitochondrial Mass, Potential, Superoxide Production, and Respiration

For the determination of mitochondrial mass, potential, and superoxide production, HUVECs were cultured, seeded into a 96-well plate, and exposed to 25% plasma derived from HF and control patients at time point 1 (baseline) and 3 (directly after cessation of CPB) as described above. Following overnight incubation, wells were washed with warm HBSS (Gibco 14025-050), and incubated for 45 min with HBSS containing 1:20,000 MitoTracker Deep Red FM (Invitrogen M22426), 1:2000 MitoTracker Green (Invitrogen M7514), and 2.5 µM MitoSOX Red (Invitrogen M36008). After incubation, wells were washed 3 times with warm EBM + 2% FCS, and incubated with EBM + 2% FCS for 45 min.

Mitochondria were imaged using max-projections of 5 z-steps with 0.7 µm step size on a high content confocal microscope (Molecular Devices, ImageXpress™ Micro Confocal) at 40× magnification (Nikon Plan Apo Lambda, Tokyo, Japan; NA = 0.95), using a 60 µm pinhole and a heated, CO_2_-enriched, and humidified sample stage. Four sites without overlap were imaged per condition.

Mitochondrial respiration was measured using the Agilent Seahorse XF Cell Mito Stress assay (Agilent, Santa Clara, CA, USA). The assay was conducted as follows: 6000 HUVECs were seeded per well, in a gelatin-coated 96-well Seahorse XF Cell Culture Microplate (Agilent, Santa Clara, CA, USA).. Cells were left to grow for 2 days in a humidified incubator at 37 °C. Cells were exposed to patient-derived plasma for 18 h as described above. Afterwards, the Seahorse XF Cell Culture Microplate was placed in the Agilent Seahorse Analyzer (Agilent, Santa Clara, CA, USA). Cells were exposed to 6 mM Glucose, 1.5 µM Oligomycin, 1 µM FCCP, and 0.5 µM Rot/AA. OCR and ECAR were measured during the course of the experiment. Results were normalized to the number of cells present in the wells.

### 2.7. Data and Statistical Analysis

For the morphological profiling, data analysis was performed in Python version 3.7.6, using the Scikit-learn package [30]. Datasets were scaled to zero-mean and unit variance, and then combined [31]. This dataset was subsequently transformed by Factor Analysis to reduce dimensions and whiten the data [32]. Single datapoints were created by averaging all cells for each patient/time point in the transformed dataset.

Baseline patient characteristics were described using summary statistics. Continuous variables were reported as mean ± SD or as median with interquartile range. Differences between groups (HFrEF vs. control) were compared using an unpaired Student’s *t*-test, or a Median test with Yates’s correction for continuity when necessary. Categorical data were reported as percentages and differences were compared based on Chi-square or Fisher’s exact test, when appropriate. To observe different trends between the two groups and mitigate the influence of the baseline differences between the two groups, the derivative and the differences between subsequent time points were calculated. One-sample *t*-tests were used to determine if delta was different from 0. The significance level was set at *p* < 0.05. Linear regression analysis was performed in R. One patient with bleeding complications was left out of the linear regression analysis. Statistical analysis was performed using SPSS (version 25.0, IBM, Armonk, NY, USA) and R software (Version 4.1.0, 2022.0201, R Foundation for Statistical Computing, Vienna, Austria).

## 3. Results

### 3.1. Patient Characteristics

In total, 36 patients were included (67 (61–71) years, 78% male) in the VASOR study (Table 1). HFrEF patients had a higher body mass index (27 ± 4 vs. 24 ± 3, *p* = 0.024). Control patients were reported to use diuretics (28% vs. 89%, *p* < 0.001) and beta-blockers (33% vs. 89%, *p* = 0.002) less often. Comorbidities were similar between the two groups. The intraoperative characteristics did not differ between control and HFrEF patients. HF patients required significantly more noradrenaline support during the postoperative course as seen by the total administered dosage (186.51 (14.77–345.03) vs. 3.61 (0–41.60) μg/kg, *p* = 0.003 in HFrEF and controls, respectively) and total duration of noradrenaline dependency (1281 (215–1615) vs. 177 (0–882) min, *p* = 0.009, in HFrEF and controls, respectively) (Appendix A).

### 3.2. Circulating Plasma Markers

The plasma concentrations of sP-Selectin, ANG2, vWF, and CD138, as measures of platelet activation, EC activation, and glycocalyx shedding, respectively, were measured for all time points (Figure 1A). The EC activation markers ANG2 and vWF showed higher concentrations at baseline for the HFrEF group—ANG-2: 6.1 ng/mL [IQR 3.1–10.6] vs. 2.7 ng/mL [IQR 2.3–4.2] ng/mL (*p* = 0.02), vWF: 2.4 U/mL [IQR 2.0–3.1] vs. 1.7 U/mL [IQR 1.4–2.3] (*p* = 0.05), for HFrEF vs. controls, respectively (Figure 1). The concentrations of both ANG2 and vWF peaked 24 h after surgery, again being higher for the HFrEF group—ANG-2: 11.8 ng/mL [IQR 8.3–17.7] vs. 7.7 ng/mL [IQR 6.4–12.3] (*p* = 0.06), vWF: 3.5 U/mL [IQR 3.0–4.0] vs. 3.3 U/mL [IQR 2.8–4.8] (*p* = 0.9), for HFrEF vs. controls, respectively.

The absolute changes in ANG-2 concentration between time points (Delta-ANG2) were similar between the two groups, and both groups showed a similar increasing trend. Delta-vWF between T4 and T3 (postoperative day 1 minus post-CPB) was two-fold higher in the non-HF patients (*p* = 0.02), even though both groups reached similar absolute vWF plasma concentrations on T4 (postoperative day 1). Importantly, delta-vWF (T4–T3) correlated with the total noradrenaline dose received in the ICU (−0.0015 ng/mL vWF/µg, 95% CI: (−0.0022:−0.00076), *p* = 0.0003, R^2^ = 0.36), and also with the duration of the noradrenaline administration (−2.09 × 10^−4^ ng/mL vWF/seconds, 95% CI: (−3.17 × 10^−4^:−1.01 × 10^−4^), *p* = 0.0004, R^2^ = 0.35) (Figure 1B,C).

The baseline sP-Selectin levels did not differ between the two groups (HF: 94 ng/mL [IQR 83–120] vs. non-HF: 78.5 ng/mL [IQR 59.5–108.5], *p* = 0.16). Interestingly, the sP-Selectin concentrations remained relatively constant throughout all time points, indicated by Delta-sP-Selectin not being significantly different from 0 for both groups and time points. The baseline CD138 was similar for the two groups (HF: 46 ng/mL [IQR 52–86] vs. non-HF: 52 ng/mL [IQR 42–66], *p* = 0.1). The CD138 concentrations peaked directly after the cessation of CPB (HF: 185 ng/mL [IQR 117–280] vs. non-HF: 150 ng/mL [IQR 105–226], *p* = 0.5). Similar CD138 trends were observed for both groups.

### 3.3. Morphological Profiling of EC Responses to Patient Plasma

The exposure of cultured ECs to patient-derived plasma resulted in visually identifiable morphological differences between the HFrEF patient and the non-HF group (Figure 2A). In addition, major changes were observed between T1 and T3 for both groups (Figure 2A). ECs exposed to plasma derived from control patients at baseline (T1) exhibited a quiescent phenotype with little F-Actin stress fibers, intact Weibel–Palade bodies, and fanned-out mitochondrial networks. ECs exposed to plasma from HFrEF patients at baseline (T1) showed more stress fibers and activation. Most importantly, the mitochondrial morphology appeared markedly different for T3 (after the cessation of CPB) for both groups, showing higher levels of fractionation.

Patient-specific EC morphological profiles, i.e., a multidimensional vector capturing the overall morphological responses of the cultured ECs to stimulation by the patient-derived plasma, were visualized in a two-dimensional UMAP (Figure 2B). Clear, discriminative clustering can be observed of the control patient baseline profiles and of the HFrEF patient baseline profiles. In addition, a clear separation between baseline (T1) and after the cessation of CPB (T3) can be observed. This separation indicates HFrEF and non-HF patient-derived plasmas induce clearly different morphologies at baseline, and that EC morphological responses are markedly changed by surgery on CPB. Furthermore, the overlap between the HFrEF patients and non-HF patients after CPB cessation (T3) indicated that surgery induced clear morphological changes resulting in similar morphologies for both groups, overruling the differences observed at baseline by factors circulating in the plasma after the cessation of CBP.

To determine which cellular components (e.g., mitochondria and cytoskeleton) contribute to the separation between the groups and time points, both factor composition (Figure 2C) and factor values per group/time point (Figure 2D) were investigated. Factor composition shows which cellular components were important for the creating of a specific factor, and factor values per group/time point indicate if there is a difference in that factor for the group or time point.

Factor 2 and 8 show a significant difference between baseline (T1) and after the cessation of CPB (T3) for both groups. These values were not different between the two groups at any of the two time points (Figure 2D). Both these factors represent mitochondrial information, as the absolute coefficients were highest for mitochondrial information (Figure 2C).

vWF/Weibel–Palade body morphology, represented by Factor 4, showed a significant difference between the HFrEF group and the control group, regardless of the time point. The F-Actin cytoskeleton morphology, VE-Cadherin border morphology, and cellular shape, represented by Factor 5 and 12, differed significantly between the two groups at baseline (T1), while differences were also apparent between baseline and post-CPB (T1–T3) for both groups. Nevertheless, after CPB cessation (T3), Factor 5 and 12 were similar in HFrEF and controls. Interestingly, Factor 6 and 7, capturing different information about the cellular shape, VE-Cadherin border morphology, and F-Actin cytoskeleton morphology, represent a change in these structures between time points, in addition to the offset of the non-HF group at baseline (T1) represented by Factor 5 and 12.

### 3.4. Mitochondrial Changes in ECs in Response to Patient Plasma

As mitochondrial morphology was a key discriminating factor between T1 (baseline) and T3 (directly after the cessation of CPB), mitochondrial function was further investigated. Mitochondrial mass, mitochondrial potential, and mitochondrial superoxide production were measured in cultured ECs exposed ex vivo to patient plasma (Figure 3A–C). The mitochondrial mass was constant between the groups and time points. Interestingly, even though mitochondrial superoxide production was not significantly elevated between time points as a group, exposure to plasma derived from certain non-HF patients after CPB cessation (T3) induced high mitochondrial superoxide production. No significant difference was observed in mitochondrial respiration (Figure 3D–H).

## 4. Discussion

In this study, we investigated the differences between EC responses to surgery on CPB for HFrEF patients and control patients. We have demonstrated that HFrEF patients exhibit preoperative EC activation and a dampened response to surgery on CPB compared to control patients. This diminished responsiveness of the native endothelial system was significantly associated with the postoperative need of noradrenaline and, consequently, with hemodynamic instability. We have shown that circulating factors are partly responsible for the activated EC phenotype of HFrEF patients, and that soluble factors released during cardiac surgery on CPB result in a highly activated EC phenotype and major morphological changes in mitochondria.

At baseline, we observed major differences in EC activation between the two groups, both in the circulating plasma markers of EC activation, as well as in the EC responses to patient-derived plasma in our in vitro endothelium model. First, HFrEF patients had significantly higher preoperative levels of vWF and ANG2, two circulating markers for endothelial activation and damage [33,34]. This indicates that HF patients have activated ECs before surgery, possibly indicating the presence of systemic endothelial dysfunction. Moreover, the exposure of ECs to baseline (T1) non-HF patient-derived plasma resulted in little activation in the cultured ECs. Conversely, ECs exposed to plasma from HFrEF patients showed higher degrees of EC activation, represented by the redistribution of vWF and the remodeling of the F-actin cytoskeleton, VE-Cadherin borders, and cellular shape. This indicates that there are certain stable, non-volatile circulating factors present in the plasma that mediate the EC activation in HFrEF patients. However, we did not identify the factors present in the plasma causing the EC responses, nor could we—due to the low sample size—identify if comorbidities such as diabetes had an effect on the baseline EC activation and responses.

We observed major differences in the EC responses during and after surgery on CPB between the two groups. The vWF levels were constant throughout surgery for the HFrEF patients, compared to a marked increase during the first postoperative day (T4) for the non-HF patients. To rule out that the increases in vWF levels were platelet-derived, we included measurements of sP-selectin, a transmembrane single-chain glycoprotein embedded on the membrane of the α-granules and often used as a platelet activation marker [35,36]. No increase in sP-selectin was observed during the course of the surgery; therefore it is unlikely that the increased vWF levels were derived from platelets, and, thus, must originate from the endothelium. Therefore, this indicates that the endothelium of HFrEF patients shows a diminished response to surgery, even though the circulating factors should elicit a similar response to the non-HF group, as indicated by the similar responses in our in vitro endothelium model for both groups.

The change in vWF levels between the cessation of CPB and postoperative day 1 (T4–T1) was significantly associated with the total duration of noradrenaline administration and the total dosage given. These findings indicate that patients with a dampened endothelial responsiveness to surgical trauma are less able to adequately maintain their vascular tone. Thus, these patients are more susceptible to postoperative hemodynamic instability and, hence, vasoplegia and necessitate prolonged vasopressor support. Our results are in accordance with a previously published study by Kortekaas et al. which reported that pre-existing EC activation is a predisposing factor for postoperative vasoplegia [37]. The pre-existing, chronic EC activation and dysregulation in HFrEF patients is, thus, an important, destabilizing factor in the pathophysiology of vasoplegia.

ANG2 levels have previously been reported to be temporally associated with endothelial glycocalyx damage, likely due to a shear-stress-mediated mechanistic link between glycocalyx injury and ANG2 expression [38]. Glycocalyx injury has been previously reported in studies in cardiac surgery patients [39,40]. We, therefore, investigated if glycocalyx injury was observed within our patient groups by measuring circulating CD138, a marker of glycocalyx degradation. This association might offer an explanation for the differences between ANG2 and vWF increases postoperatively. Indeed, CD138 levels peaked after CPB in both groups without significant differences, highlighting the universal impact of surgical trauma and CPB on endothelial glycocalyx regardless of HF status. Therefore, the differences between ANG2 and vWF measurements likely arise from the added effect of glycocalyx injury on ANG2 levels, which is similar for the two groups.

Interestingly, our in vitro endothelium model showed very similar responses to patient-derived plasmas after CPB cessation (T3) for both non-HF and HFrEF patients. This indicates that, in naïve, cultured ECs, the responses to surgery on CPB are the same between the two groups. In addition to high levels of EC activation, the mitochondrial morphology was also vastly affected by the patient-derived plasma post-surgery, resulting in mitochondrial fission and fragmentation. Stable, circulating plasma factors must be responsible for this effect. Likely, reactive oxygen species products from reperfusion injury, DAMPs released by surgical trauma, and hemin products from hemolysis contribute to the EC and mitochondrial morphology we observed as a result of the exposure to patient-derived plasma collected after CPB cessation (T3) [41]. Unfortunately, these DAMPs and hemolytic products can only be removed by plasmapheresis and, therefore, are an unlikely candidate for intervention.

This study has several limitations that should be considered. First of all, the HF group had more patients suffering from diabetes, a condition that also contributes to endothelial damage. Unfortunately, the HbA1c values preoperatively were not available and we cannot control for this. Future studies should consider a more balanced distribution of important comorbidities like diabetes in order to better delineate the influence of HF on endothelial function. Furthermore, the ex vivo stimulation does not account for pre-existing endothelial cell and mitochondrial damage in the HFrEF group as HUVECs are used as a model. Having access to cultured endothelial cells from patients would allow us to study the effects of surgery on CPB in an even more relevant scenario, and prove whether ROS-exposed mitochondria of HFrEF are indeed more damaged by the circulating plasma factors. However, the use of HUVECs was chosen as a method to investigate the reaction of endothelial cell cultures to plasma contents that are released in the circulation of patients with and without HF in response to cardiac surgery on CPB. It certainly does not completely reflect the patients’ endothelial cell responses, but it gives an insight on the effects on the endothelial function of certain factors that are released. Last, this is an analysis of a small, single-center study. This limited us to stratifying for different levels of heart failure, etiology, and the time since the HF onset. These factors might influence the post-operative course of the individual patient and cannot be corrected for within the current study. Therefore, larger patient cohorts are necessary in order to confirm its results and further investigate possible associations.

## 5. Conclusions

Pre-existing EC dysfunction in HFrEF patients likely accounts for a less robust endothelial response to surgical stress. This seems to be responsible for the susceptibility of these patients to postoperative hemodynamic instability and vasoplegia. Furthermore, the distinctive mitochondrial morphological changes after surgery indicate that the already stressed mitochondria in the endothelium of HFrEF patients are likely at an increased risk of dysfunction. Further research is guaranteed to investigate the involvement of endothelial dysregulation and mitochondrial dysfunction in HFrEF patient in the development of postoperative hemodynamic instability and vasoplegia. In addition, future research should focus on exploring the plasma factors that trigger this endothelial and mitochondrial damage. Such factors could possibly help identify patients at risk for postoperative hemodynamic instability and aid the development of preventive measures.

## Figures and Tables

**Figure 1 cells-14-01357-f001:**
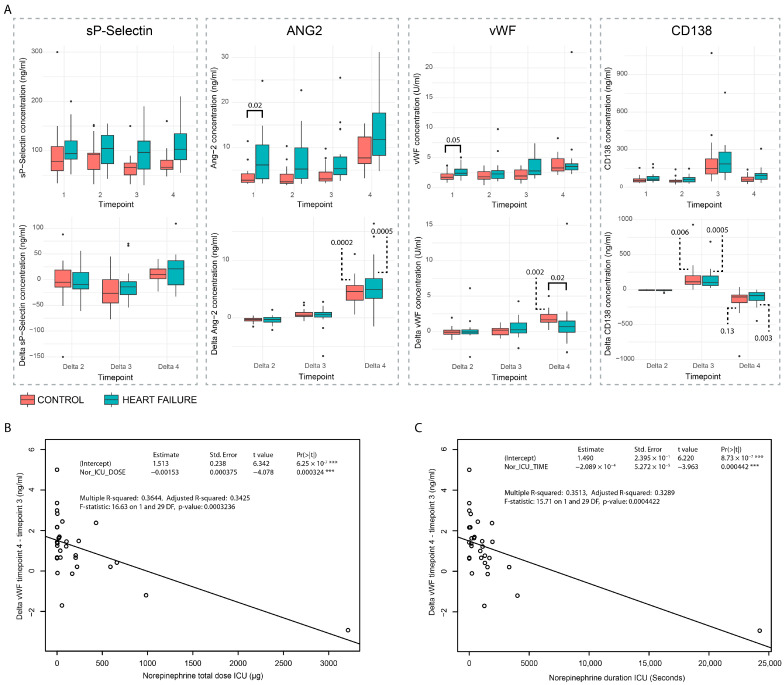
(**A**) Circulating markers of EC activation/damage in patient-derived plasma. ANG2: angiopoetin-2, CD138: syndecan-1, sP-Selectin: soluble P-Selectin, vWF: von Willebrand factor. Delta 2: Change in levels between before and after anesthesia induction (T2–T1), Delta 3: Change in levels between after anesthesia induction and cessation of cardiopulmonary bypass (T3–T2), Delta 4: Change in levels between cessation of cardiopulmonary bypass and postoperative day one (T4–T3). Brackets indicate comparisons between two groups; dotted lines indicate significance compared to the condition of Delta = 0. For each time point, *n* = 18 HFrEF and *n* = 18 control patients were included. Unpaired *t*-tests were used to compare between the different groups. For delta-values, one-sample *t*-tests were used. (**B**) Linear regression model of total noradrenaline dose received in the ICU versus changes in vWF between time point 3 and time point 4. *n* = 35 patients were included, and one patient was excluded due to bleeding complications. (**C**) Linear regression model of total noradrenaline duration in the ICU versus changes in vWF between time point 3 and time point 4. *n* = 35 patients were included; one patient was excluded due to bleeding complications.

**Figure 2 cells-14-01357-f002:**
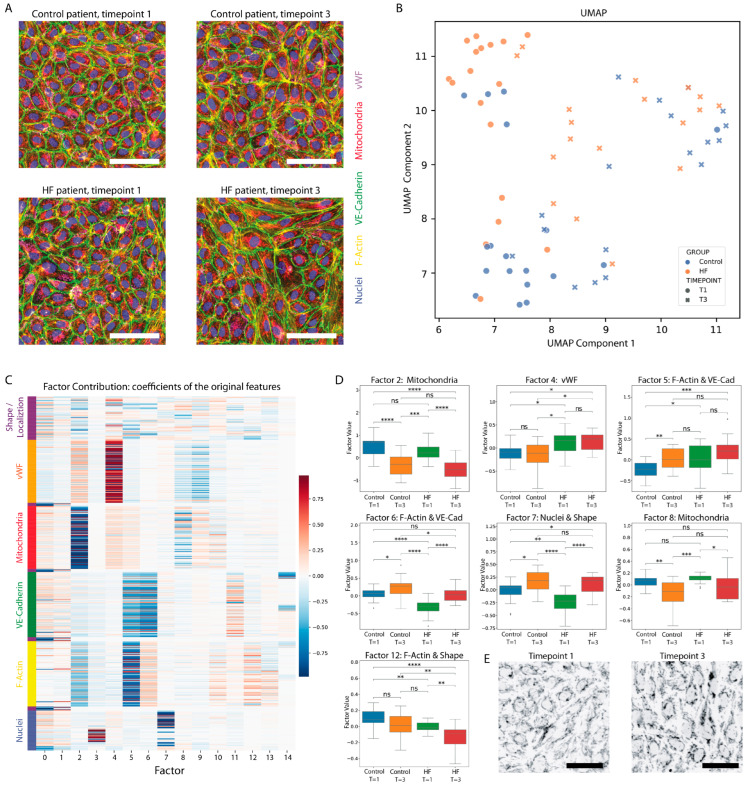
Endothelial cell morphological profiling. (**A**) Examples of EC morphology after stimulation with patient plasma for time point 1 and 3 for both a control patient and HF patient. ECs exposed to HF patient plasma look more activated in general, as can be observed from the F-actin stress fibers. At time point 3, activation and mitochondrial morphological changes can be observed for both groups. Scalebar = 100 µm. (**B**) UMAP of the EC morphological profiles for each individual patient–time point. At baseline (round dots), a clear separation of the control and HF can be observed. Moreover, separation between time point 1 and time point 3 can be observed, where HF and control patient morphological profiles are more similar at time point 3. Per patients, features were extracted from over 4000 cells and combined into the patient profile after dimension reduction. For time point 1, *n* = 18 control and *n* = 18 HFrEF patients were included, and, for time point 3, *n* = 17 control and *n* = 18 HFrEF were included. (**C**) Composition of the factors based on the coefficient of the original feature. A higher intensity color indicates a larger absolute coefficient, and, hence, a higher contribution to that factor. Original features are grouped based on the cell component they are derived from; i.e., nucleus, F-Actin cytoskeleton, etc. For time point 1, *n* = 18 control and *n* = 18 HFrEF patients were included, and, for time point 3, *n* = 17 control and *n* = 18 HFrEF were included. (**D**) Factor values for factors that showed at least one significant difference between the groups/time points were plotted for group and time point. The presented data represents 18 patients per group for two time points. Non-parametric Mann–Whitney test was used. For time point 1, *n* = 18 control and *n* = 18 HFrEF patients were included, and, for time point 3, *n* = 17 control and *n* = 18 HFrEF were included. (**E**) Example of mitochondrial morphology for time point 1 and 3. Scalebar = 50 µm. *: 0.05 > *p* > 0.01. **: 0.01 > *p* > 0.001. *** 0.001 > *p*. **** 0.0001 > *p*. ns: non significant.

**Figure 3 cells-14-01357-f003:**
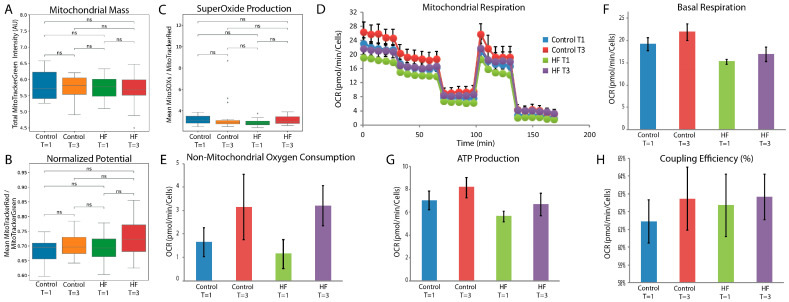
Endothelial cell mitochondrial mass, potential, and superoxide production. (**A**) Mitochondrial mass as measured by total masked MitoTracker green signal showed no difference between all groups and time points. 250 cells were scored per patient, for *n* = 18 patients per group, with Mann–Whitney U-test. (**B**) Mitochondrial potential normalized by mitochondrial mass showed no significant difference between the groups and time points. 250 cells were scored per patient, for *n* = 18 patients per group, with Mann–Whitney U-test. (**C**) Superoxide production normalized for mitochondrial potential showed that some patients in the control group produce high levels of superoxide at time point 3. 250 cells were scored per patient, for *n* = 18 patients per group, with Mann–Whitney U-test. (**D**) Oxygen Consumption Rate (OCR) during the course of the experiment. (**E**) Non-mitochondrial oxygen consumption after adding of Antimycin-A and Rotenone. (**F**) Mitochondrial oxygen consumption at baseline after adding Glucose. (**G**) ATP production as measured by the drop in OCR after adding Oligomycin. (**H**) Coupling efficiency. NS: Not significant.

**Table 1 cells-14-01357-t001:** Baseline and historic disease characteristics of patients in the VASOR study (*n* = 36).

	Heart Failure (*n* = 18)	Control (*n* = 18)	*p* Value
Age (years) (IQR)	68 (62–71)	64 (59–69)	0.279
Male sex (%)	67	89	0.228
Body mass index (kg/m^2^) (±SD)	27 ± 4	24 ± 3	**0.024**
Diabetes (%)	28	6	0.177
Prior hypertension (%)	39	17	0.264
Pulmonary hypertension (%)	28	11	0.402
Previous cardiac surgery (%)	22	6	0.338
Hemoglobin (mmol/L) (±SD)	8.5 ± 1.0	9.1 ± 1.0	0.093
Creatinine clearance (mL/min/1.73 m^2^) (±SD)	67 ± 20	78 ± 16	0.090
EuroSCORE II (%) (IQR)	9.76 (6.59–15.49)	1.45 (1.07–2.71)	**<0.001**
Left ventricular ejection fraction (%) (±SD)	24.9 ± 6.4	60.3 ± 6.9	**<0.001**
**Medication use**			
Beta blocker (%)	89	33	**0.002**
Angiotensin-converting-enzyme inhibitor/angiotensin receptor blockers (%)	61	50	0.738
Antiarrhythmics (%)	28	11	0.402
Mineralocorticoid receptor antagonists (%)	44	11	0.060
Diuretics (%)	89	28	**<0.001**
Inotropes (%)	11	0	0.486
**Procedure type**			
Mitral valve plasty (%)	56	94	**0.018**
Tricuspid valve plasty (%)	33	50	0.500
Surgical left ventricular restoration (%)	39	0	**0.008**
Left ventricular assist device implantation (%)	22	0	0.104
Coronary artery bypass grafting (%)	39	6	**0.041**
Aortic valve replacement (%)	17	0	0.229
Aorta surgery (%)	11	6	1.000

Note: Data presented as mean ± SD, median with interquartile range (IQR), or percentage for count data. *p* Values in bold are statistically significant. Fisher exact test was used for count data, independent Student *t*-test was used for data presented as mean ± SD, and Median Mann–Whitney U test was used for data presented as median with interquartile range (IQR).

## Data Availability

All relevant data is presented in the manuscript. Raw data is available upon request from the corresponding author.

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
