# Peer review of "Heart Failure Impacts Endothelial Cell Responses to Cardiac Surgery on Cardiopulmonary Bypass"

_cells, 2025, doi:10.3390/cells14171357_

Round 1

Reviewer 1 Report

Comments and Suggestions for Authors

I wish to congratulate with the author for the very well arranged and clearly presented research and manuscript.

One major limitation that I see in the study design is that is not clearly stratified the level of HF, the medication type, and the time since the HF onset. Can be understood from the type of surgery performed that etiology of HF is different, and degree as well, given that some patient are going to heart reparative surgery, other to heart replacement therapy.

Furthermore cardiopulmonary bypass duration, as well as heart disease etiology leading to cardiac surgery might differently affect EC performance.

This has to be stated for the different impact that might have on the research outcome, as is well know on the post operative course.

Author Response

Dear reviewer,

We thank you for your valuable remarks and kind words.

We were indeed limited with the smaller sample size in exploring these combined factors. We have addressed your limitation in the 'limitations of the study' section, by including the sentence: "this is an analysis of a small, single center study. This limited us to stratify for different level of heart failure, etiology, and the time since the HF onset. These factors might influence the post-operative course of the individual patient and cannot be corrected for within the current study. Therefore, larger patient cohorts are necessary in order to confirm the results and further investigate possible associations" 

We hope including this is satisfactory and thank you again for your valuable remarks.

Best,

Rudmer Postma

Reviewer 2 Report

Comments and Suggestions for Authors

Dear Authors: 

you investigated a highly relevant topic in regard to endothelial cell responses to cardiac surgery on cardiopulmonary bypass. : This study is based on the VASOR study. 
Blood samples from HFrEF and control patients undergoing cardiac surgery on CPB were collected (pre- and perioperatively @4 time points). Levels of Angiopoietin 2, von Willebrand Factor and soluble P-selectin  were measured by ELISA. Additionally, EC responses to plasma were evaluated via morphological profiling and mitochondrial functional assays. 36 predominantly male patients were investigated. It has been highlighted, that HFrEF patients exhibit a dampened EC response to cardiac surgery on CPB. Stable circulating factors in HFrEF plasma are responsible for inducing EC stress. Moreover, mitochondrial function is highly affected postoperatively. The preexisting mitochondrial and EC dysfunction predispose HFrEF patients to postoperative hemodynamic instability.

Fantastic paper with excellent methodology. Congratulations.

Comments on the Quality of English Language

English language is excellent.

Author Response

Dear Reviewer,

We thank you for your kind words and appreciation of our manuscript!

Best wishes on behalf of all authors,

Rudmer Postma

Reviewer 3 Report

Comments and Suggestions for Authors

In the study the authors evaluated differences between EC responses to surgery on CPB for HFrEF patients and control patients. They found that  HFrEF patients had impaired response to surgery on CPB compared to control patients. The manuscript is well organised, clearly written and contains important scientific information that is undoubtedly of high practical value.Despite the impressive results of the study, I would like to make some suggestions for discussion around them.

  1. It remains unclear what causes endothelial progenitor dysfunction in patients with HFrEF and how likely it is that the comorubid profile may be a determining factor in the impaired EC response.
  2. The authors concluded that "circulating factors are partly responsible for the activated EC phenotype of HFrEF patients, and that soluble factors released during cardiac surgery on CPB result in a highly activated EC phenotype and major morphological changes in mitochondria". It remaines uncertain whether the impaired activity of EC may affect proliferative ability, differentiation capacity through mitochondrial alteration.
  3. Which of the factors activating the immune system could be targeted by therapy, and to what extent could this approach be of practical value?

Author Response

Dear Reviewer,

We thank you for your kind words and valuable comments!

Indeed, we were not able to investigate exactly which factor in the plasma is causing these changes, nor do we have the needed sample size to investigate if a combination of comorbidities influences the EC responses. We addressed comment 1 by including the following sentence in the manuscript: "However, we didn’t identify the factors present in the plasma causing the EC responses, nor could we -due to the low sample size- identify if comorbidities such as diabetes had an effect the baseline EC activation and responses."

2: Although it would be interesting to investigate the effect of HFrEF and especially mitochondrial dysfunction on EC proliferation, our focus for this manuscript is on the more immediate effects on EC dysfunction that are potentially at the base of post-operative complications that occur soon after surgery. Therefore we chose to only focus on non-proliferative, contact-inhibited, ECs.

3: We appreciate your focus on actionability of our findings. Likely the EC dysfunction and the mitochondrial phenotype is caused by DAMPs and hemolytic products in the plasma. Unfortunately, the way to remove these products is by plasmapheresis after surgery, thus vastly complicating the surgery. We addressed this comment by adding the following sentence: "Likely, reactive oxygen species products from reperfusion injury, DAMPs released by surgical trauma and hemin products from hemolysis contribute to the EC and mitochondrial morphology we observed as a result of exposure to patient-derived plasma collected after CPB cessation (T3). [37] Unfortunately, these DAMPs and hemolytic products can only be removed by plasmapheresis and therefore are an unlikely candidate for intervention."

On behalf of all authors, we hope we addressed your comments satisfactory and thank you again for your valuable input!

Best,

Rudmer Postma